# Effects of Intermittent Hypoxia on Pulmonary Vascular and Systemic Diseases

**DOI:** 10.3390/ijerph16173101

**Published:** 2019-08-26

**Authors:** Hiroshi Kimura, Hiroyo Ota, Yuya Kimura, Shin Takasawa

**Affiliations:** 1Department of Advanced Medicine for Pulmonary Circulation and Respiratory Failure, Graduate School of Medicine, Nippon Medical School, Bunkyo, Tokyo 113-8603, Japan; 2Department of Respiratory Medicine, Nara Medical University, Kashihara, Nara 634-8522, Japan; 3Center for Pulmonary Diseases, NHO Tokyo National Hospital, Kiyose, Tokyo 204-0023, Japan; 4Department of Biochemistry, Nara Medical University, Kashihara, Nara 634-8521, Japan

**Keywords:** sleep apnea, intermittent hypoxia, sympathetic nerve, pulmonary hypertension, REM sleep, lifestyle-related diseases, insulin secretion, insulin resistance

## Abstract

Obstructive sleep apnea (OSA) causes many systemic disorders via mechanisms related to sympathetic nerve activation, systemic inflammation, and oxidative stress. OSA typically shows repeated sleep apnea followed by hyperventilation, which results in intermittent hypoxia (IH). IH is associated with an increase in sympathetic activity, which is a well-known pathophysiological mechanism in hypertension and insulin resistance. In this review, we show the basic and clinical significance of IH from the viewpoint of not only systemic regulatory mechanisms focusing on pulmonary circulation, but also cellular mechanisms causing lifestyle-related diseases. First, we demonstrate how IH influences pulmonary circulation to cause pulmonary hypertension during sleep in association with sleep state-specific change in OSA. We also clarify how nocturnal IH activates circulating monocytes to accelerate the infiltration ability to vascular wall in OSA. Finally, the effects of IH on insulin secretion and insulin resistance are elucidated by using an in vitro chamber system that can mimic and manipulate IH. The obtained data implies that glucose-induced insulin secretion (GIS) in pancreatic β cells is significantly attenuated by IH, and that IH increases selenoprotein P, which is one of the hepatokines, as well as TNF-α, CCL-2, and resistin, members of adipokines, to induce insulin resistance via direct cellular mechanisms. Clinical and experimental findings concerning IH give us productive new knowledge of how lifestyle-related diseases and pulmonary hypertension develop during sleep.

## 1. Introduction

Obstructive sleep apnea (OSA) is common in the general population, and its prevalence rate ranges from 9 to 38% [1]. OSA causes a variety of systemic disorders pertaining to lifestyle-related diseases through mechanisms such as sympathetic nerve activation, systemic inflammation, and oxidative stress [2,3]. Polysomnographic recording in OSA shows intermittent hypoxia (IH), clinically determined as recurrent swings of arterial oxygen saturation (SpO_2_) by pulse oximeter, which is associated with repeated sleep apnea followed by snoring and hyperventilation. It is clinically important to resolve and clarify pathophysiological mechanisms caused by IH.

Systemic effects of nocturnal IH also impinge on pulmonary circulation. Besides hypoxic pulmonary vasoconstriction (HPV) induced by alveolar hypoxia in OSA, nocturnal pulmonary vascular tone was elucidated to be modified by additional mechanisms dependent on sleep state- specific changes, as described below [4,5,6]. In the daytime condition, OSA patients usually show augmented respiratory drives to compensate for hypoventilation via peripheral chemoreceptors, such as carotid body (CB) as well as central respiratory control mechanisms [4,5]. Additionally, central load compensation mechanisms in response to the increases in upper airway resistance and chest wall mechanical load are concerned with this protective action to maintain arterial carbon dioxide level eucapnia [6]. During sleep, however, these protective mechanisms are generally impaired because of atonia in both the postural muscles and the upper airway dilating muscles, and ventilation is only maintained by diaphragmatic activity and central drive [7]. Patients with the hypoventilation phenotype, typically with obesity hypoventilation syndrome (OHS), fail to maintain their respiratory drives during sleep and this can eventually lead to daytime hypercapnia and hypoventilation [8].

Right heart failure and pulmonary hypertension (PH) are not necessarily common in typical OSA patients without hypoventilation [9]. In OHS, however, PH is observed in up to 50% of patients [10]. Shirai and his colleagues demonstrated that pulmonary sympathetic nerve activities are modulated in response to systemic hypoxia in a rat model [11]. Moreover, they clarified that pulmonary vascular tone depends on the central modulation of sympathetic nerve activity during IH [12].

## 2. Effects of Intermittent Hypoxia on Pulmonary Vascular Diseases

Excitation and/or instability of sympathetic nerve activities are also characteristic of OSA associated with repeated apnea [13]. Nocturnal IH causes recurrent and dramatic increases in pulmonary arterial pressure (PAP) in OSA (Figure 1). This increase in PAP and intermittent PH occurs in different manners between rapid eye movement (REM) sleep and non-REM (NREM) sleep [14]. Pulmonary vascular tone can be elevated in accordance with alveolar hypoxia, which elicits hypoxic pulmonary vasoconstriction (HPV) in both REM and NREM sleep. We reported that additional mechanisms are concerned with the increase in PAP during REM sleep, which is independent of the degree of hypoxia [14]. Although alveolar hypoxia is generally recognized to cause HPV, state-specific change impinging on pulmonary vasculature was independent of HPV. Surprisingly, this REM sleep-specific increase in PAP was observed in accordance with phasic REM events even under nasal continuous positive airway pressure (CPAP) treatment with SpO_2_ maintained at more than 90% (Figure 2). It is suggested that IH associated with sympathetic nerve activation during REM sleep plays an important role in causing daytime PH in OSA.

Recent findings suggest that the increase in sympathetic activity is generally known as a pathophysiological mechanism in hypertension [15] and insulin resistance [16,17]. Insulin resistance is broadly observed in patients with visceral obesity, and it was reported to be a common feature in OSA whether or not obesity is associated [18,19,20,21,22,23]. The afferent discharges of both CB and renal sympathetic nerves are augmented in hypertensive patients, and both discharges respond to hypoxia. Besides their sensing function as chemoreceptors to hypoxia, CBs also play an important role in controlling metabolic homeostasis as glucose sensors. Ribeiro and his colleagues claimed that the overactivation of CB function by hyperglycemic diet induced insulin resistance [24]. It is conceivable that both the overactivation of sympathetic nervous systems and insulin resistance influence each other.

A cross-sectional cohort study showed that sleep disordered breathing (SDB) is an independent risk factor of coronary heart disease, heart failure, and stroke. The prevalence of these cardiovascular diseases in subjects with SDB was 1.3 to 2.4 times compared to the control [25]. Nocturnal IH is associated with metabolic risk factors after the adjustment of various confounding factors, such as age, sex, and body weight. Importantly, these characteristics were reported to not necessarily be influenced by body weight increase, as they were more remarkable in non-overweight subjects rather than overweight ones [26].

Repetitive hypoxia and reoxygenation and the accompanying shear stress to vascular beds potentially cause sympathetic activation, hypoxia-reoxygenation injury, and many inflammatory mechanisms [27]. Hypoxic stress causes not only inflammation via monocyte/macrophage (Mφ) activation, but also the migration of monocytes/Mφ into the vascular wall [28]. This process results in atherosclerosis [29]. Furthermore, OSA is associated with the increase in oxidative stress or antioxidant deficiencies, which is involved in the activation of redox-sensitive transcription factors [30]. Endothelial dysfunction is clarified to be caused by oxidative stress, which results in cardiovascular diseases. Suzuki and his colleagues evaluated carotid-artery intima-media thickness (IMT) with ultrasonography in OSA [31]. They clearly showed for the first time that the more apnea hypoxia index (AHI), the more the IMT increases, and this relationship was independent of age.

Tamaki et al. demonstrated that invasion ability of monocytes can be evaluated by a Matrigel invasion chamber system. Monocyte suspension taken from the subject’s blood was added to the upper chamber, oxidized low density lipoprotein (LDL) was added to the lower chamber as chemoattractant [32]. They were cultured for 24 h, and invasive cells in the lower membrane were counted under the microscope. The number of invasive cells can be considered to represent the invasion ability of monocytes. The numbers of invasive monocytes were much higher in OSA patients compared to control subjects. Furthermore, in OSA patients, monocytes invasiveness was significantly augmented in the early morning after waking up, suggesting that nocturnal hypoxic stress activates monocytes/Mφ. Surprisingly, this activated function of monocytes/Mφ rapidly recovered to the normal level following treatment with CPAP in OSA.

Insulin resistance in type 2 diabetes is broadly recognized to be associated with obesity. Common genetic variants associated with obesity in adults have been identified by genome-wide association studies [33]. Analysis of genetic variants revealed that an increase in adult body mass index (BMI) is associated with genetic factors until the age of 20 years, but that environmental factors rather than genetics are involved after this age [33]. These findings suggest that nocturnal IH might be an important etiological factor to develop type 2 diabetes via the influence of recurrent environmental changes in adults. The association between nocturnal IH and the risk of development of type 2 diabetes has been reported in community-based studies [34,35,36]. The hazard ratio for developing type 2 diabetes was 1.69 with a 3% oxygen desaturation index (3% ODI) of more than 15.0 compared with a 3% ODI <5.0.

## 3. Effects of Intermittent Hypoxia on Systemic Diseases

There were no studies concerning the direct effects of IH on pancreatic β cells from the viewpoint of cellular mechanisms until that reported by us in 2012 [37]. Pancreatic β cells were exposed either to normoxia or to IH, which mimics the environmental condition of OSA patients, for 5 min hypoxia of 1% O_2_ followed by 10 min normoxia for 24 h using a custom-designed, computer-controlled incubation chamber system. In the normoxic condition, glucose-induced insulin secretion (GIS) was observed in β cells. After IH treatment, GIS was significantly attenuated compared to that in the normoxic condition [37]. Glucose signaling to insulin secretion is initiated by the uptake of glucose and the subsequent metabolism of the sugar in β cells is essential to insulin secretion. In this respect, we examined the messenger RNA (mRNA) levels of insulin and several genes involved in Ca^2+^ influx from extracellular sources using rodent pancreatic β cells. The level of insulin mRNAs (Insulin 1 and Insulin 2) as well as mRNAs for the Ca^2+^ influx from extracellular sources (glucose transporter 2, glucokinase (d-hexose 6-phosphotransferase), sulfonylurea receptor 1, and l-type Ca^2+^ channel 1.2) were not changed by IH. Next, the mRNA level of CD38 (ADP-ribosyl cyclase/cyclic ADP-ribose hydrolase), which is a key enzyme acting as a second messenger for intracellular Ca^2+^ mobilization in the CD38-cyclic ADP-ribose signal system [38,39], was significantly lower in IH-treated cells than in normoxia-treated cells. This indicates that IH stress directly attenuates GIS from β cells via the downregulation of CD38. Furthermore, Ota et al. demonstrated that β cell numbers increased in IH, due to the fact that IH stress stimulates pancreatic β cells to induce interleukin-6 (IL-6) gene expression. This in turn induces the overexpression of Reg family genes [40] and the hepatocyte growth factor gene [41], which leads to an increase the numbers of β cells [42] (Figure 3). The occurrence(s) of autoantibodies and single-nucleotide polymorphisms (SNPs) of human CD38 and REG Iα have been reported in diabetes patients [43,44,45,46]. Therefore, OSA patients with such antibodies and/or SNPs could be susceptible to type 2 diabetes. In fact, autoantibodies against CD38 were reported in Graves’ disease [47] and lupus erythematosus patients [48], and autoantibodies against REG protein were detected in Japanese Sjögren’s syndrome patients [49].

The insulin resistance of peripheral tissues as well as the impairment of glucose-induced insulin secretion from pancreatic β cells causes type 2 diabetes. To evaluate the effect of IH on insulin resistance, Uchiyama et al. studied the expression of hepatokines in the condition of IH using cultured human hepatocytes and the in vitro system as described above. They found that selenoprotein P was upregulated in hepatocytes by IH [50]. Selenoprotein P is one of the hepatokines synthesized and secreted from hepatocytes that has been established to induce insulin resistance [51]. The effects of IH on selenoprotein P gene expression showed that IH increased selenoprotein P expression via the upregulation of selenoprotein P mRNA in human hepatocytes, although other hepatokines, such as α2 HS-glycoprotein, angiopoietin-related growth factor 6, epidermal growth factor 21, leukocyte cell-derived chemotaxin 2, lipasin, and sex hormone-binding globulin, were not increased by IH [50]. They concluded that IH stress upregulates the levels of selenoprotein P in human hepatocytes to accelerate insulin resistance. It also upregulates the mRNA levels of the hepatocyte growth factor hepatocarcinoma-intestine-pancreas/pancreatitis-associated protein (HIP/PAP) [40,52] to proliferate such hepatocytes (Figure 4). The upregulation of these two mRNAs was revealed to occur via a microRNA-203-mediated mechanism [50].

The worldwide prevalence of OSA is increasing due to its close association with obesity and multiple health complications, such as hypertension, cardiovascular disease, and type 2 diabetes [53]. Angiopoietin-like protein (ANGPTL)4 and ANGPTL8 have been suggested to play a role in the development of these diseases through their role in regulating the metabolism of plasma lipid molecules [54]. Abubaker and his colleagues demonstrated that ANGPTL4 and 8 levels were increased in subjects with OSA, suggesting that the upregulation of these lipid metabolism regulators might play a role in the lipid dysregulation observed in patients with OSA [54]. ANGPTL4 and 8 are expressed and secreted mainly from liver and adipose tissues. It could be important to identify from where the increased ANGPTL4 and 8 were derived—liver and/or adipose tissues. Obesity induces insulin resistance in peripheral insulin target tissues, and adipose tissue is considered to play a central role in insulin resistance. Most recently, Uchiyama and her coworkers exposed mouse 3T3-L1 and human SW972 adipocytes to IH and analyzed the expression of adipokines (IL-6, adiponectin, leptin, tumor necrosis factor-α (TNF-α), C-C motif chemokine ligand 2 (CCL2), and resisin). They found that the mRNA levels of TNF-α, CCL2, and resistin were significantly increased in response to IH and that the increases were caused by the IH-induced decrease of microRNA-452 in hepatocytes [55]. As TNF-α plays a central role in obesity-related insulin resistance, CCL2 is a key player in the development and maintenance of chronic adipose tissue inflammation and insulin resistance, and resistin is associated with insulin resistance, the IH-induced upregulation of these adipokines in adipocytes induces and/or worsens insulin resistance and/or type 2 diabetes in OSA patients (Figure 5).

Accumulating evidence indicates that obesity and OSA are strongly related to each other [56,57]. However, a prospective nonrandomized controlled study revealed that body mass index (BMI) was significantly lower in OSA Far East Asian men compared to OSA white men when controlled for sex, age, and disease severity; in fact, the mean BMI of Far East Asian men with OSA was below the norm for men in the United States [58]. Therefore, the mechanism by which OSA affects patients’ body weight remains unclear. Although the etiology of obesity is complex and energy balance is regulated by many neurobiological and physiological mechanisms, weight gain is generally supposed to result from excessive food intake leading to an imbalance between calorie intake and energy expenditure. The effect of IH on the regulation of appetite and feeding behavior in OSA patients has been obscure. However, no reports have examined the changes in the expression of appetite regulatory genes under the influence of IH. Recently, Shobatake and his colleagues reported that IH upregulated the mRNA levels of the anorexigenic peptides proopiomelanocortin (POMC) and cocaine- and amphetamine-regulated transcript (CART) but not those of galanin, galanin-like peptide, ghrelin, pyroglutamylated RFamide peptide, agouti-related peptide, neuropeptide Y, and melanocortin 4 receptor in human neuronal cells [59]. This indicates that IH inhibits appetite and food intake in OSA patients by increasing mRNAs for POMC and CART in the central nervous system (CNS). Appetite and food intake are controlled by not only the CNS but also by the gastrointestinal (GI) tract, both of which work together as the gut–brain axis, representing a bidirectional signaling axis [60]. Gut peptides, which are released from enteroendocrine cells within the epithelium throughout the GI tract, might activate vagal and spinal afferents indirectly via the activation of neurons of the enteric nervous system (ENS) and relay nutrient-derived energy signals to the brain, so that appetite and food intake could be regulated appropriately through the gut–brain axis [60]. Shobatake and colleagues thus hypothesized that IH could have an anorexigenic effect on the ENS, in addition to the CNS. They investigated the effect of IH on the gene expression(s) of major appetite-inhibiting gut peptide hormones, and found that peptide YY, glucagon-like peptide-1, and neurotensin were upregulated in human enteroendocrine cells by IH [61].

Today, metabolic diseases significantly contribute to early death in Western society. More than 422 million people worldwide are estimated to have diabetes, causing 3% of global deaths [62]. Metabolic diseases also contribute to one-third of all cancers [63] and cancer-related deaths [64]. Type 2 diabetes is a metabolic disease that is associated with obesity, reduced insulin-stimulated glucose uptake by skeletal muscle and adipose tissue, and impaired β cell function [65]. Skeletal muscle is responsible for the majority of insulin-sensitive glucose uptake. In vivo studies present conflicting data. Some suggest that IH induces insulin resistance, while some show improvements in insulin sensitivity. A study using a mouse model of IH showed not only decreases in whole-body insulin sensitivity, but also reduced glucose utilization and insulin sensitivity in the soleus muscle, suggesting a clear reduction in glucose metabolism and uptake into this muscle. The impact of IH was most pronounced in oxidative muscle fibers (soleus muscle), while glycolytic muscle (vastus lateralis muscle) and mixed oxidative and glycolytic (gastrocnemius muscle) fibers were relatively unaffected. Thus, glucose uptake in oxidative muscle tissue is significantly impaired by IH and this effect appears independent of obesity [66]. On the other hand, Mackenzie and colleagues showed that acute hypoxic exposure increased insulin sensitivity in individuals with type 2 diabetes [67]. These findings were confirmed by Lecoultre et al., who showed that 10 nights of moderate hypoxic exposure improved insulin sensitivity in obese males [68]. The question of whether hypoxia causes insulin resistance or not is a complex one. There are few studies that have examined the effect of IH on skeletal muscle glucose uptake and metabolism. Recently, it was found that muscle cells express and secrete several cytokines, called myokines [69]. As some myokines—such as IL-8, osteonectin, and myonectin—are involved in inflammation and glucose metabolism, IH could modulate the expression of such myokines. Therefore, myokines could be new targets for future research on IH and/or IH-related diseases.

OSA is also a risk factor for cardiovascular diseases (e.g., atherosclerosis) as well as type 2 diabetes [70,71]. IH could cause not only pulmonary vascular disease but also systemic vascular diseases. In fact, Kyotani and his colleagues recently demonstrated that the epidermal growth factor family, such as epiregulin, amphiregulin, and neuregulin-1, and their receptor erbB2 were upregulated in vascular smooth muscle cells by IH [72], and these upregulations were mediated by the increase of IL-6 [73,74]. These findings indicate that IH seen in OSA patients modulates not only pulmonary but also systemic vascular pathophysiology.

The results of the abovementioned studies explain that IH causes several pathophysiological phenomena such as decreased glucose-induced insulin secretion and increased insulin resistance via direct cellular mechanisms.

## 4. Conclusions

Intermittent hypoxia during sleep is frequently observed in a high number of OSA patients. Clinical and experimental findings derived from these patients give us important and new knowledge concerning how lifestyle-related diseases and pulmonary hypertension develops and how they are associated with intermittent hypoxia. Further translational studies are required to resolve these issues and to develop treatment strategies.

## Figures and Tables

**Figure 1 ijerph-16-03101-f001:**
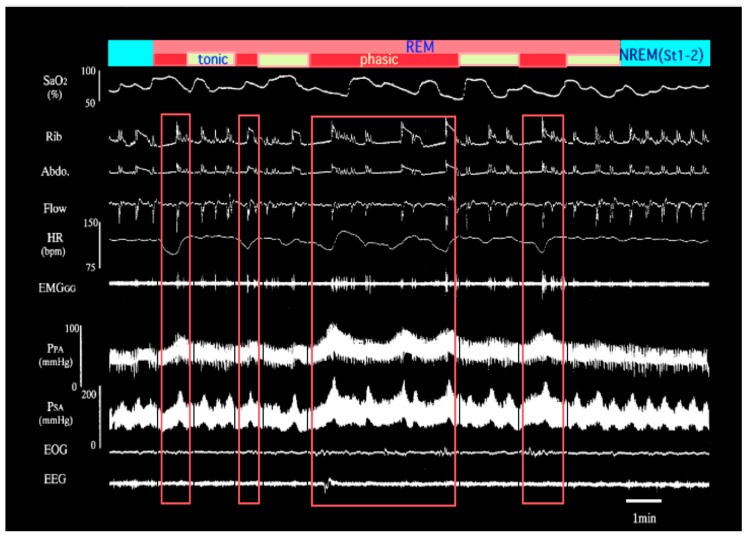
Representative polysomnographic recording during transition from non-REM (NREM) to REM to NREM sleep in an obstructive sleep apnea (OSA) patient with daytime pulmonary hypertension (PH) (adapted from [14]). Increase in pulmonary artery pressure is more exaggerated during REM sleep than NREM sleep. Moreover, the rapid rise of pulmonary arterial pressure (PAP) associated with the appearance of phasic REM and its recovery to the initial level immediately after the disappearance of REM are evident. The record of the area surrounded by the red square lines represents REM sleep. SpO_2_; arterial oxygen saturation by pulse oximeter, Rib; rib cage movement, Abdo; abdominal movement, Flow; nasal airflow, HR; heart rate, EMG_GG_; genioglossal electromyogram, P_PA_; pulmonary artery pressure, P_SA_; systemic artery pressure, EOG; electrooculogram, EEG; electroencephalogram. Phasic; phasic REM, tonic; tonic REM.

**Figure 2 ijerph-16-03101-f002:**
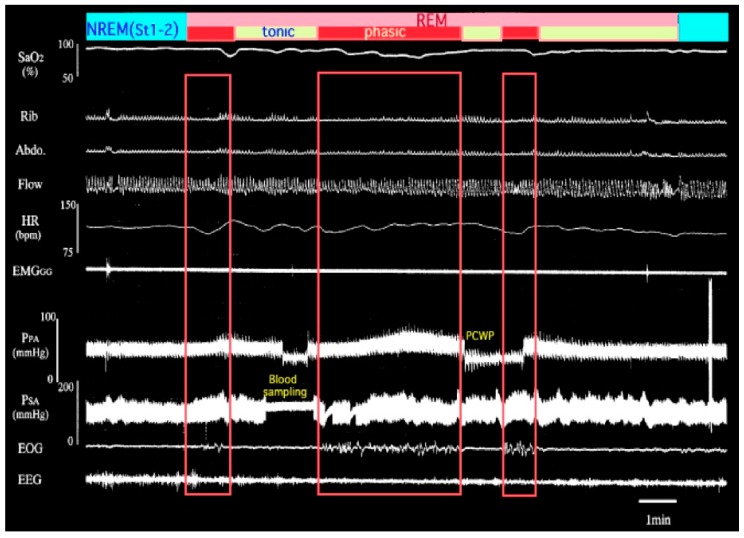
Polysomnographic recording in the same patient who was treated with nasal continuous positive airway pressure (CPAP). Tremendous desaturation before CPAP treatment was restored. However, the increase in pulmonary artery pressure can be observed in association with the appearance of phasic REM, suggesting the participation of neural control. REM-specific elevation in PAP occurred independently of the degree of hypoxia. PCWP; pulmonary capillary wedge pressure. SpO_2_; arterial oxygen saturation by pulse oximeter, Rib; rib cage movement, Abdo; abdominal movement, Flow; nasal airflow, HR; heart rate, EMG_GG_; genioglossal electromyogram, P_SA_; systemic artery pressure, EOG; electrooculogram, EEG; electroencephalogram. Phasic; phasic REM, tonic; tonic REM.

**Figure 3 ijerph-16-03101-f003:**
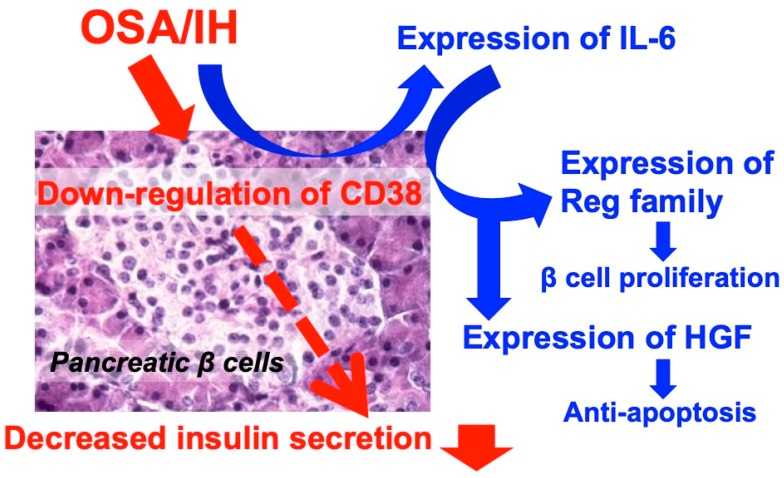
Possible effects of intermittent hypoxia (IH) on glucose-induced insulin secretion and β cell proliferation. IH causes the downregulation of CD38 to decrease glucose-induced insulin secretion, as well as the upregulation of IL-6. IL-6 increases the expression of Reg family genes and HGF gene Reg family members function as growth factors for pancreatic β cells and HGF works as an antiapoptotic factor. As a result, pancreatic β cells with decreased glucose-induced insulin secretion are increased.

**Figure 4 ijerph-16-03101-f004:**
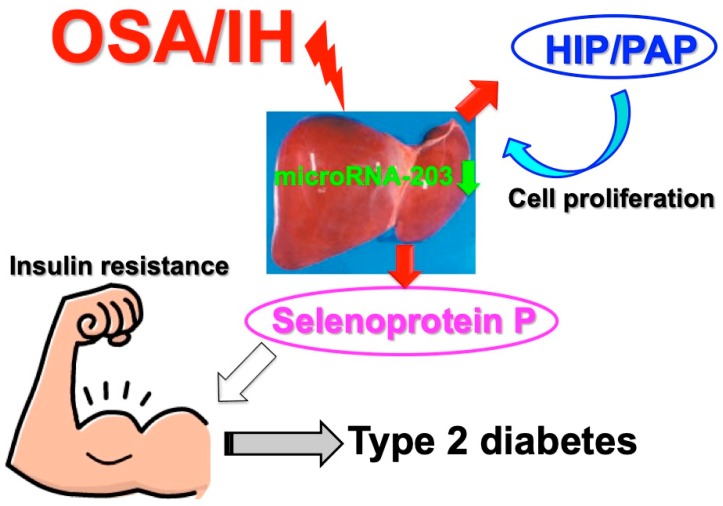
A possible mechanism of IH-induced insulin resistance by selenoprotein P. IH stress downregulates microRNA-203 in hepatocytes. The mRNAs for selenoprotein P and HIP/PAP target microRNA-203. As a result, IH exposure upregulates hepatokine(s) such as selenoprotein P to increase insulin resistance, as well as HIP/PAP to increase hepatocyte proliferation [50].

**Figure 5 ijerph-16-03101-f005:**
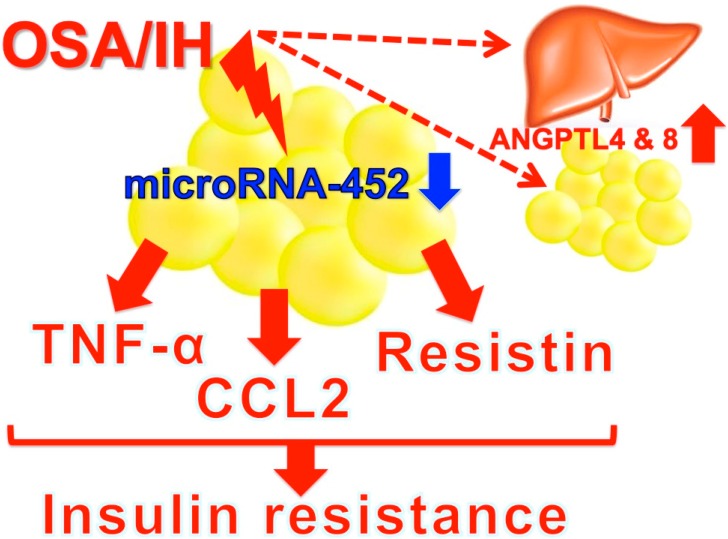
A possible mechanism of IH-induced insulin resistance. IH stress upregulates serum levels of ANGPTL4 and 8 [54] as well as the expression of adipokine(s) such as TNF-α, CCL2, and resistin via the downregulation of microRNA-452 to increase insulin resistance [55]. Secreted ANGPTL4, ANGPTL8, TNF-α, CCL2, and resistin all work together to lead to insulin resistance and/or type 2 diabetes in OSA patients.

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
