# Peer review of "Effects of Intermittent Hypoxia on Pulmonary Vascular and Systemic Diseases"

_ijerph, 2019, doi:10.3390/ijerph16173101_

Round 1
Reviewer 1 Report
The paper by Kimura et al. aims to describes the effects of Intermittent Hypoxia on Pulmonary Vascular and Systemic Diseases. This paper is clear but needs some corrections.
Major comments:
Line 33: The prevalence rate is very old. In the last two or three years has been published many reports of prevalence, including systematic reviews. Please update these low rates. For example, 9% to 38% in the adult population (Senaratna, 2017).
Line 54: the authors say, “PH is occasionally observed up to 50% of patients”. If the percentage is up to 50%, is not occasionally.
Line 94-96: It is possible to add prevalence or other epidemiological data of this fact?
Line 109: It is interesting the explanation of hypoxia and re-oxygenation, but the authors do not use the term endothelial dysfunction that has been reported extensively in this pathology. I think that the authors need to discuss this paragraph in term of endothelial dysfunction produced by the hypoxia and reoxygenation phenomena.
Line 247-249: In the sentence: “On the other hand, Mackenzie and coleagues showed that [54] acute hypoxic exposure increased insulin sensitivity in individuals with Type 2 diabetes, and ten nights of moderate hypoxic exposure improved insulin sensitivity in obese males [55].” Yo cite two articles, but only write one idea. Please, check the reference 55 in this sentence.
Minor comments:
Line 43: Please add the reference for this sentence
Line 50: please add the reference
Line 52: please add the reference
Line 60: please add the reference
The authors use pulmonary arterial pressure several times. Please use abbreviation PAP and add in the listo f abbreviations.
Line 72: Is OSA or OAS?
Line 111, 112 and 113: please add the reference
Line 185: add the reference
Line 187: add the reference
Line 225: add the reference
Line 229: add the reference
Line 247: change coleagues by colleagues
Author Response
To Reviewer 1.
Thank you very much for your fruitful comments and advice.
According to your comments, we changed the manuscript as a revised version.
Major comments:
Concerning Line 33 in the ORIGINAL VERSION;
According to reviewer’s advice, we changed the description concerning prevalence rate to a recent report, including systematic reviews as follows;
“Obstructive sleep apnea (OSA) is common in the general population, and its prevalence rate ranged from 9% to 38% [1].” in the REVISED VERSION (line 34).
Reference 1 was also changed as follows:
Senaratna, C.V.; Perret, J.L.; Lodge, C.J.; Lowe, A.J.; Campbell, B.E.; Matheson, M.C.; Hamilton, G.S.; Dharmage, S.C. Prevalence of obstructive sleep apnea in the general population: A systematic review. Sleep Med. Rev. 2017, 34, 70-81. in the REVISED VERSION.
Concerning Line 54 in the ORIGINAL VERSION;
According to reviewer’s advice, we deleted the term “occasionally”, and changed as follows;
In OHS, however, PH is observed up to 50% of patients. in the REVISED VERSION (lines 55-56).
Line 94-96: It is possible to add prevalence or other epidemiological data of this fact?
According to the kind suggestion, we added some epidemiological references in the REVISED VERSION (Jehan, S.; Myers, A.K.; Zizi, F.; Pandi-Perumal, S.R.; Jean-Louis, G.; McFarlane, S.I. Obesity, obstructive sleep apnea and type 2 diabetes mellitus: Epidemiology and pathophysiologic insights. Sleep Med. Disord. 2018, 2, 52-58.; Seetho, I.W.; Wilding, J.P. Sleep-disordered breathing, type 2 diabetes and the metabolic syndrome. Chron. Respir. Dis. 2014, 11, 257-275.; Bonsignore, M.R.; Esquinas, C.; Barceló, A.; Sanchez-de-la-Torre, M.; Paternó, A.; Duran-Cantolla, J.; Marín, J.M.; Barbé, F. Metabolic syndrome, insulin resistance and sleepiness in real-life obstructive sleep apnoea. Eur. Respir. J. 2012, 39, 1136-1143.) as Ref. 20-23.
Line 109
Thank you for Reviewer’s valuable comment. (1) According to the advice, we rewrite the sentence as follows.
Repetitive hypoxia and re-oxygenation, and accompanying shear stress to vascular beds, potentially cause sympathetic activation, hypoxia-reoxygenation injury, and a lot of inflammatory mechanisms. Hypoxic stress causes not only inflammation via monocytes/ macrophages (Mφ) activation, but also migration of monocytes/Mφ into vascular wall. These process results in atherosclerosis. Furthermore, OSA is associated with the increase in oxidative stress or antioxidant deficiencies which is involved in the activation of radox-sensitive transcription factors [30]. Endothelial dysfunction is clarified to be caused by oxidative stress, which results in cardiovascular diseases. Suzuki T et al. evaluated carotid-artery intima-media thickness (IMT) with ultrasonography in OSA [31]. They clearly showed in the first time that the more AHI is, the more IMT increased, and this relationship was independent of their ages.
We added a new reference (Ref. 30 in the REVISED VERSION)
Yamauchi, M.; Kimura, H. Oxidative stress in obstructive sleep apnea: putative pathways to the cardiovascular complications. Antioxid. Redox Signal. 2008, 10, 755-768.
Line 247-249: In the sentence: “On the other hand, Mackenzie and coleagues showed that [54] acute hypoxic exposure increased insulin sensitivity in individuals with Type 2 diabetes, and ten nights of moderate hypoxic exposure improved insulin sensitivity in obese males [55].” You cite two articles, but only write one idea. Please, check the reference 55 in this sentence.
According to the Reviewer #1’s kind suggestion, we changed the sentence “On the other hand, Mackenzie and colleagues showed that [54] acute hypoxic exposure increased insulin sensitivity in individuals with Type 2 diabetes, and ten nights of moderate hypoxic exposure improved insulin sensitivity in obese males [55].” To “On the other hand, Mackenzie and colleagues showed that acute hypoxic exposure increased insulin sensitivity in individuals with Type 2 diabetes [67], and the findings were confirmed by Lecoultre et al. that ten nights of moderate hypoxic exposure improved insulin sensitivity in obese males [68].”
Minor comments:
Line 43: Please add the reference for this sentence
According to the Reviewer #1’s suggestion, we added references [4-6] in the REVISED VERSION.
Line 50: please add the reference
According to the Reviewer #1’s suggestion, we added a reference [7] in the REVISED VERSION.
Line 52: please add the reference
According to the Reviewer #1’s suggestion, we added a reference [8] in the REVISED VERSION.
Line 60: please add the reference
According to the Reviewer #1’s suggestion, we added a reference [13] in the REVISED VERSION.
The authors use pulmonary arterial pressure several times. Please use abbreviation PAP and add in the list of abbreviations.
According to the Reviewer #1’s kind suggestion, we used abbreviation “PAP” as pulmonary arterial pressure and add “PAP” in the list of abbreviations.
Line 72: Is OSA or OAS?
Thank you for kind advice. “OSA” is correct. We change “OAS” (line 72 in the ORIGINAL VERSION) was changed to “OSA” in the REVISED VERSION (line 72).
Line 111, 112 and 113: please add the reference
According to the Reviewer #1’s suggestion, we added references [27, 28, and 29] in the REVISED VERSION.
Line 185: add the reference
According to the Reviewer #1’s suggestion, we added a reference [53] in the REVISED VERSION.
Line 187: add the reference
According to the Reviewer #1’s suggestion, we added a reference [54] in the REVISED VERSION.
Line 225: add the reference
According to the Reviewer #1’s suggestion, we added a reference [60] in the REVISED VERSION.
Line 229: add the reference
According to the Reviewer #1’s suggestion, we added a reference [60] in the REVISED VERSION.
Line 247: change coleagues by colleagues
As the Reviewer 2 pointed out, “coleagues” (line 247 in the ORIGINAL VERSION) was changed to “colleagues” in the REVISED VERSION (line 256).

Reviewer 2 Report
The review article titled “Effects of Intermittent Hypoxia on Pulmonary Vascular and Systemic Diseases” seems quite interesting and relevant. I liked the flow and information presented in article. The article is well constructed and offers the great correlation related to the obstructive sleep apnea (OSA), intermittent hypoxia (IH), and systematic diseases. Finally, I recommend that the paper should be accepted for the publication after minor revision (See comments below) depending upon the discretion of other reviewer’s.
Line 24 should read “pancreatic β cells were significantly” Lines 103-104 were needs to be re-written, readability is poor. Line 114 should read “They clearly showed for the first time that the more” If possible, please resize the figures and make sure colors are more pleasing? Lines 209-212 are quite complexly written, If possible, simplify the sentence? Any recurrent hypothesis, how pathogenesis of OSA is increasing with hypertension, cardiovascular disease, and Type 2 diabetes as stated in lines 183-184?Overall, the paper is properly written and much readable in general with substantial amount of information. If the relevant comments will be addressed the article will have a great impact in understanding the pathogenesis of OSA and IH induced diseases and physiological conditions. In my opinion, this paper needs minor revision before publishing with journal based on recommendations of other reviewers.
Good Luck!!!
Author Response
To Reviewer 2.
Thank you very much for your valuable comments.
In response to your comments, we sincerely reply as follows.
Line 24,
We carefully checked the sentence, and we would like to mention that “glucose-induced insulin secretion (GIS) in pancreatic β cells was significantly attenuated by IH,” does not need a correction.
Line 103-104,
We checked the sentence, and we would like to mention that “Cross-sectional cohort study shows that sleep disordered breathing (SDB) is independent risk factors of coronary heart disease, heart failure and stroke. And the prevalence of these cardiovascular diseases in subjects with SDB was 1.3 to 2.4 times comparing to control [25].” does not need a correction.
Line 114,
According to Reviewer’s advice, we changed the sentence as follows;
They clearly showed for the first time that the more AHI is, the more--.
On the comments about “resize the figures and make sure color are more pleasing”;
Although we understand the reviewer’s comment is reasonable, I am sorry to say that the original figure we can keep with us is only these figures. As Figure 1 is published in the American Review of Respiratory Disease in 1999, it is not possible to prepare another figures rather than Figure 1 and Figure 2. So we would like to use these important figures.

Round 2
Reviewer 1 Report
After reviewing the manuscript, I think it meets the standards to be published.